# Tobramycin Supplemented Small-Diameter Vascular Grafts for Local Antibiotic Delivery: A Preliminary Formulation Study

**DOI:** 10.3390/ijms222413557

**Published:** 2021-12-17

**Authors:** Mariella Rosalia, Priusha Ravipati, Pietro Grisoli, Rossella Dorati, Ida Genta, Enrica Chiesa, Giovanna Bruni, Bice Conti

**Affiliations:** 1Department of Drug Sciences, Pharmaceutical Section, University of Pavia, Via Taramelli 12, 27100 Pavia, Italy; mariella.rosalia01@universitadipavia.it (M.R.); r.priusha@gmail.com (P.R.); rossella.dorati@unipv.it (R.D.); ida.genta@unipv.it (I.G.); 2Department of Drug Sciences, Pharmacological Section, University of Pavia, Via Taramelli 16, 27100 Pavia, Italy; 3Department of Surgery, Fondazione IRCCS Policlinico San Matteo, 27100 Pavia, Italy; enrica.chiesa@unipv.it; 4Consorzio per lo Sviluppo dei Sistemi a Grande Interfase (C.S.G.I.) & Department of Chemistry, Physical Chemistry Section, University of Pavia, Via Taramelli 10, 27100 Pavia, Italy; giovanna.bruni@unipv.it

**Keywords:** local drug delivery, small-diameter vascular graft, electrospinning, microbicidal effect, tobramycin

## Abstract

Peripheral artery occlusive disease is an emerging cardiovascular disease characterized by the blockage of blood vessels in the limbs and is associated with dysfunction, gangrene, amputation, and a high mortality risk. Possible treatments involve by-pass surgery using autologous vessel grafts, because of the lack of suitable synthetic small-diameter vascular prosthesis. One to five percent of patients experience vascular graft infection, with a high risk of haemorrhage, spreading of the infection, amputation and even death. In this work, an infection-proof vascular graft prototype was designed and manufactured by electrospinning 12.5% *w*/*v* poly-L-lactic-co-glycolic acid solution in 75% *v*/*v* dichloromethane, 23.8% *v*/*v* dimethylformamide and 1.2% *v*/*v* water, loaded with 0.2% *w*/*w*_PLGA_. Polymer and tobramycin concentrations were selected after viscosity and surface tension and after HPLC-UV encapsulation efficiency (EE%) evaluation, respectively. The final drug-loaded prototype had an EE% of 95.58% ± 3.14%, with smooth fibres in the nanometer range and good porosity; graft wall thickness was 291 ± 20.82 μm and its internal diameter was 2.61 ± 0.05 mm. The graft’s antimicrobic activity evaluation through time-kill assays demonstrated a significant and strong antibacterial activity over 5 days against *Staphylococcus aureus* and *Escherichia coli*. An indirect cell viability assay on Normal Human Dermal Fibroblasts (NHDF) confirmed the cytocompatibility of the grafts.

## 1. Introduction

Peripheral artery occlusive disease (PAOD) is an emerging cardiovascular disease, consisting of the blockage of blood vessels in legs or arms causing reduced blood flow to the concerned tissues. PAOD is primarily connected to the formation of atherosclerotic plaques, which are a major cause of death with a predicted global annual mortality of 23.3 million by 2030 [1,2]. Approximately 12% to 20% of people aged over 60 develop peripheral arterial disease, which is often associated with extensive limb pain, poor quality of life and a high risk of amputation and death. Even though upper extremity thrombosis/emboli occurs less frequently than in lower extremities, it constitutes approximately 15% to 20% of all emboli, and approximately 70% are of cardiac origin [3]. Developing surgical treatment options for upper extremity vascular complications is of high significance as they are often diagnosed very late; moreover, due to the lack of functional small diameter grafts (diameter < 6 mm), it is challenging to treat microvascular defects or disorders. Current treatment strategies include performing balloon angioplasty, inserting stents or performing bypass surgery. Current gold-standard vascular grafts for bypass surgeries are autologous saphenous vein, internal mammary artery and internal thoracic artery, but their use is often limited due to unavailability or pre-existing unfit autologous grafts and often fail because of complications due to mismatch of compliance, low patency, thrombosis and intimal hyperplasia, requiring further reintervention [4,5]. To overcome these disadvantages, the development of high performance readily available synthetic grafts is required. Currently, expanded polytetrafluoroethylene (ePTFE) and polyethylene terephthalate (Dacron) are the only clinically used vascular grafts. These grafts are non-degradable and are often used for bypass graft surgeries to replace medium or large-sized blood vessels. When it comes to replacing smaller blood vessels with diameters less than 6 mm, they are prone to occlude due to thrombosis and compliance mismatch, having lower patency rates compared to autologous veins or arteries, and due to the risk of infections [6]. 

Moreover, approximately 1–5% of patients develop infections after implantation of vascular prosthetics, even though the surgical procedure is categorized as clean surgery by the National Research Council. Surgical wound infections are very dangerous, with an increased risk of amputations estimated at 4–14% or even mortality with a rate of 10–25% within 30 days of diagnosis [7]. The most common Gram-positive bacteria causing infections are *Staphylococci*, including *S. aureus*, which is the most prevalent microorganism, especially in early infections, methicillin-resistant *S. aureus* and *S. epidermidis*, followed by Gram-negative microorganisms such as *Escherichia coli* and *Pseudomonas aeruginosa*; other possible causative bacteria belong to the genus *Streptococcus*, *Enterococcus* and *Enterobacter* [7]. Most of those bacteria can form biofilms, which is a sessile biocenosis of microorganisms irreversibly attached to a surface and embedded in self-produced extracellular matrix. The bacteria growing in this community are protected against environmental stresses, from the immune system and from disinfectants and antibiotics, due to their restricted penetration into the thick biofilm [8,9]. Once the vascular graft is infected and the biofilm formed, bacteria can easily spread to the native vessels, leading to inflammation and erosion of the anastomosis, thereby causing haemorrhage and formation of false aneurysm. Thus, although prosthetic vascular graft infections are a rare complication of vascular reconstruction surgery, they are associated with significant morbidity and mortality. Vascular graft infections are often treated by using intravenous narrow or wide-spectrum antibiotics such as daptomycin, linezolid and vancomycin against Gram-positive bacteria, β-lactams and fluoroquinolones for Gram-negative bacteria and especially *Pseudomonas* species [8]. Other adjunctive treatments include antibiotic loaded beads, such as with vancomycin, tobramycin, daptomycin or gentamicin, depending on the infection causative microbe [10]. But in care standards a more aggressive management strategy is preferred, including total excision of the graft, debridement of the infected tissue and extraanatomical bypass if collateral circulation is inadequate, leading to failure of the vascular reconstructive surgery in the patient [8].

The aim of this work was to design and manufacture an infection-proof and bioresorbable small-diameter (2.5 mm) vascular graft prototype using the electrospinning technique. Poly-L-lactic-co-glycolic acid (PLGA) was used because it is a biocompatible and biodegradable polyester that gives non-toxic degradation products and is FDA approved for tissue engineering and drug delivery applications [11,12,13,14,15]. In this preliminary work, attention is focused on mimicking those features of extracellular matrix and radial artery in terms of fibre diameter, pore size and matrix porosity. Through electrospinning, it is possible to obtain small fibres in the order of micro- and nano-meters with a high surface area that can effectively mimic the extracellular matrix and guide cell attachment and proliferation. Moreover, good porosity can be achieved, allowing cell infiltration into the scaffold, along with waste and nutrient exchange [6,11,16,17]. The main focus of this preliminary work concerns some aspects that, according to the literature, can be challenging, i.e., the ability to obtain an electrospun tubular small diameter graft exerting bactericidal action for a prolonged time. Therefore, electrospinning and solution parameters were set according to the final desired vascular graft features, such as wall thickness, fibre diameter, pore size and porosity. Tobramycin was selected as the antimicrobial drug because it is a wide-spectrum antibiotic that is effective against both the Gram-positive and Gram-negative aerobic bacteria involved in vascular graft infections. Moreover, it is characterized by a fast bactericidal effect, even against multidrug resistant microorganisms, and its use in vascular surgery is raising interest [10,18]. The drug was added in the bulk polymer solution, and fibres were directly spun on a rotating mandrel [14], obtaining a tubular drug-loaded graft in only a few steps. The graft was then characterized for its morphology and microbicidal efficiency. 

## 2. Results

### 2.1. Polymer Solution Characterisation

In a preliminary study, 12.5% and 15% *w*/*v* PLGA solutions in 75% *w*/*v* DCM: 23.8% *w*/*v* DMF: 1.2% *w*/*v* water with 0.01% *w*/*v* Na_2_S_2_O_5_, were evaluated for their dynamic viscosity and surface tension. In Table 1, viscosity at 100 s^−1^ and surface tension values at 25 °C and 35 °C of the polymer solutions are reported. Viscosities were taken at 100 s^−1^, because it corresponds to the shear stress applied to the polymer solution during extrusion through the syringe [19].

No significant differences in surface tension resulted among the different temperatures and polymer concentrations tested. Figure 1 shows the differences in viscosity between the PLGA solutions tested at the two set temperatures. A significant increase in viscosity occurred between the two polymer solutions depending on polymer concentrations. Concerning the effect of temperature, it was possible to observe a significant (*p* < 0.01) reduction in viscosity of 12.5% polymer solutions when temperature was increased from 25 °C to 35 °C, while no significant difference was observed for 15% PLGA solutions at the two different temperatures.

### 2.2. Tobramycin Loading in Electrospun PLGA Fibres

Tobramycin was directly added to the polymer solution in order to encapsulate the drug into the electrospun fibres. The drug encapsulation efficiency was calculated, starting from the total drug content of grafts prepared from 12.5% *w*/*v* PLGA solutions with two different tobramycin concentration, namely 0.2% *w*/*w* and 0.4% *w*/*w* with respect to PLGA. As reported in Figure 2, a 95.58% ± 3.14% encapsulation efficiency was achieved by loading 0.2% *w*/*w*_PLGA_ tobramycin into the polymer solution, whereas the 0.4% *w*/*w*_PLGA_ drug-loaded solutions gave fibres with only a 24.53% ± 5.13% encapsulation efficiency, which corresponded to a drug loading of 1.913 ± 0.064 μg/mg and 0.983 ± 0.212 μg/mg, respectively. This means that tobramycin loading into polymer grafts is highly affected by its solubility.

### 2.3. Electrospun Small-Diameter Vascular Grafts Characterisation

Placebo grafts and tobramycin loaded grafts were electrospun from 12.5% *w*/*v* PLGA solutions, with 75% *w*/*v* DCM: 23.8% *w*/*v* DMF: 1.2% *w*/*v* Na_2_S_2_O_5_ water solution (final concentration of Na_2_S_2_O_5_ in polymer solution was 0.01% *w*/*v*) as solvent system, adding 0.2% *w*/*w*_PLGA_ of tobramycin to obtain antibiotic loaded fibres. The electrospinning parameters were set up by performing a preliminary screening. The same electrospinning parameters were applied for both grafts. The obtained placebo and tobramycin loaded grafts had an internal diameter of 2.62 ± 0.07 mm and 2.61 ± 0.05 mm, and a wall thickness of 291.95 ± 32.64 μm and 291 ± 20.82 μm, respectively (see Figure 3). Scanning electron microscopy imaging allowed the morphometric characterization of the electrospun fibres, by mean of the ImageJ software. In Figure 4, SEM images of both the inner and outer surface of the placebo (PL-graft) and tobramycin loaded grafts (TB0.2-graft) are shown, highlighting their morphological differences. On the inner surface of both graft types, the fibres were random coiled, with an average diameter of 1.13 ± 0.22 μm for PL-grafts and 1.0 ± 0.15 μm for TB0.2-grafts; only 29% of the PL-graft inner layer fibres had a diameter less than 1 μm, whereas in the TB0.2-graft 57% of the fibres were in the nanoscale. The placebo grafts had a slightly higher porosity (49%) on the inner surface than drug loaded grafts (43%); the mean pore diameter for PL-grafts was 4.53 ± 5.00 μm, with 34% of the pores having a diameter greater than 5 μm, whereas in TB0.2-grafts the inner layer had pores with a mean diameter of 4.36 ± 10.25 μm average diameter, with 17% of those having a size greater than 5 μm. On the outer surface of the grafts, the fibres were smaller, with an average diameter of 0.587 ± 0.150 μm for PL-grafts and 0.893 ± 0.188 μm for TB0.2-grafts; up to 98% of the placebo fibres and 73% of the drug loaded fibres were smaller than 1 μm. It is also possible to observe a loss of orientation of the fibres on the outer surface of the TB0.2-grafts in comparison to the outer surface of the PL-grafts. The outer porosity of both types of grafts was 49%, with PL-grafts having pores with a mean diameter of 5.20 ± 7.12, from which 34% were greater than 5 μm, whereas TB0.2-grafts had a mean pore diameter of 6.29 ± 7.90 μm, with 40% of the pores having a diameter greater than 5 μm.

### 2.4. Antimicrobial Time-Kill Test

The antimicrobic activity of the TB0.2-grafts was tested for 120 h (5 days) against two bacterial strains, *Staphylococcus aureus* and *Escherichia coli*, using PL-grafts as a positive control for bacterial growth. Figure 5 clearly shows that *S. aureus* and *E. coli* populations remain almost constant within the testing period when placed in contact with PL-grafts. In contrast, there was an evident reduction of the microbial populations in contact with the antibiotic loaded graft, in the order of magnitude of a 1,000,000-fold and 100,000,000-fold decrease within the first 48 h, for *S. aureus* and *E. coli*, respectively (see Table 2).

After 48 h of contact, the TB0.2-grafts showed a constant microbicidal effect of approximately 8-log reduction for both bacterial strains until the end of the fifth day.

### 2.5. Cell Viability after Contact with Vascular Graft Extracts

Preliminary cytotoxicity evaluation on the PL and TB0.2 extracts was carried out over the same time span used for antimicrobial activity evaluation, in order to acquire comparable data. The rationale was to evaluate possible cytotoxicity which could compromise graft safety, hence the proposed use of the graft as a drug delivery system. Both placebo- and tobramycin-loaded grafts were tested for their cytotoxicity. Extracts of the grafts were prepared by incubating either placebo or tobramycin loaded graft samples in DMEM for 24, 48, 72, 96 and 120 h. All extracts were checked for their pH, which resulted as being between 7.4 and 7.5, indicating there was no massive release of PLGA soluble degradation products such as monomer or oligomers. The extracts were then incubated for 24 h with Normal Human Dermal Fibroblasts (NHDF), and cell viability was tested using MTT assay. Phenol was used as a negative cell viability control. No decrease in cell viability could be detected, either for PL-grafts or for TB0.2-grafts, after comparison with the control (see Figure 6), whereas the negative control gave 0% cell viability.

## 3. Discussion

Electrospinning is a process in which micro- and nanofibers of a polymer can be obtained when the polymer solution is subjected to an external electric field. Usually, an electric potential difference is maintained between the tip of the solution filled syringe and the collector. This applied voltage at the syringe tip causes the transfer of charge and charge repulsion in the polymer solution, resulting in stretching and the formation of a Taylor cone. When the accumulated charge in the solution is large enough to overcome the surface tension of the solution, the Taylor cone stretches into continuous fibres with simultaneous evaporation of the solvent and deposition on the oppositely charged collector. Material parameters such as polymer concentration and solvent type can affect the surface tension, viscosity and conductivity of the polymer solutions and thus their electro spinnability. Moreover, temperature can influence both viscosity and surface tension and represents an important factor in the electrospinning process affecting fibre elongation and drying. For a solution to be electro spinnable, a minimum polymer concentration is needed in order to obtain an optimal viscosity for sufficient polymer chain entanglement making continuous fibres, and a low enough surface tension is needed to allow easier ejection of the polymer solution from the syringe [20,21,22,23]. Two different polymer solution compositions were studied for their viscosity and surface tension at room temperature (25 °C) and at 35 °C. The higher temperature was chosen according to the final temperature range in which electrospinning was performed (37.4 °C–44.8 °C) and could not be overcome during viscosity and surface tension analysis because of the formation of bubbles in the solution due to the rapid evaporation of DCM (T_boil_ = 40 °C), which interfered with the measurements. This phenomenon did not occur during electrospinning because the solution was subjected to a higher pressure inside the syringe than the atmospheric pressure. DCM and DMF were used as solvent systems because they have a low surface tension of 28.20 dyne/cm and 36.42 dyne/cm, respectively, and they are therefore suitable to allow the formation of polymer fibres instead of beads, when the PLGA solution is pumped through the needle and subjected to an electric field [24]. Moreover, DMF has a high dielectric constant (ε = 36.7 [25]) that enables the stretching of the forming fibres. The polymer solution surface tension data shown in Table 1 are in accordance with the surface tension values of the pure solvents. No significant change in the surface tension of the polymer solution occurred after increasing polymer concentrations or temperature because of the restricted ranges tested for both concentrations and temperatures. Concerning the viscosity data, as expected, a higher concentration of polymer caused an increase in viscosity at both 25 °C and 35 °C, because of the higher amount of polymer molecules and hence of chain entanglements [26], and this at both temperatures. An increase in temperature caused a decrease in viscosity by enhancing polymer chain mobility and was significant only for less concentrated polymer solutions (12.5% *w*/*v* PLGA), whereas at higher concentrations (15% *w*/*v*) the influence of temperature on viscosity was less pronounced. The result is in keeping with the lower mobility of polymer chains in a higher concentrated solution. Preliminary electrospinning trials were performed with both solutions in order to evaluate their ability to be electrospun: 12.5% *w*/*v* solution had a stable Taylor cone, with good polymer jet whipping and a widespread distribution of the fibres on the collector, and a low cleaning frequency of the needle tip was required, even if a small needle gauge was used. On the other hand, 15% *w*/*v* PLGA solution was not easy to spin, with an unstable Taylor cone and frequent interruption of the polymer jet due to clogging of the needle. In fact, the viscosity of 15% PLGA solution was greater than 2000 mPa*s, which is considered the higher limit of viscosity for the electro-spinnability of polymer solutions; stronger chain entanglements occur at higher polymer concentrations, causing flow instability of the polymer solution and hence impeding the formation of a stable Taylor cone [24]. For these reasons, 12.5% *w*/*v* PLGA solutions was selected to proceed with drug loading and the manufacturing of PL- and TB0.2-grafts. 

Preliminary study on drug encapsulation in the vascular grafts showed a very low drug content of 0.983 μg/mg of the graft when loaded with 0.4% tobramycin *w*/*w* of PLGA polymer, resulting in only 24.53 ± 5.13% encapsulation efficiency. This lower encapsulation efficiency can be explained due to the inhomogeneous suspension of tobramycin in the PLGA polymer solution at higher concentrations, due to the very low solubility of the drug in the solvent blend. Moreover, frequent clogging of the syringe needle during electrospinning was observed, thus requiring more regular cleaning, leading to drug loss. On the other hand, the PLGA polymer solution with a lower concentration of 0.2% tobramycin *w*/*w* of PLGA formed a homogenous suspension in the polymer solution. Drug content analysis on the grafts electrospun with polymer solution containing 0.2% *w*/*w* tobramycin showed a 1.9-fold increase in drug content in comparison to 0.4% *w*/*w* tobramycin grafts and showed 95.6 ± 3.14% encapsulation efficiency. These results indicate there is an upper loading limit for drugs that are not soluble in the polymer solution. Similar to other aminoglycosides, tobramycin has dose-dependent side effects: vestibular ototoxicity, which causes even permanently balance disorders, and reversible nephrotoxicity, characterized by renal impairment and an increase of creatinine, urea and other metabolic byproducts in blood plasma. Less common is neuromuscular blockade, which leads to respiratory depression [18]. By reducing the loaded amount of tobramycin and by using a local drug delivery approach, these possible systemic side effect could be reduced or even avoided, meanwhile maintaining an effective antimicrobial effect.

The electrospun grafts were designed to have morphologic properties mimicking those of an autologous small diameter artery (radial artery), with fibre properties close to those of the extracellular matrix. In its final in vivo application, the graft is intended to be implanted in the human body in order to replace small diameter arteries. The rationale is that the graft should support tissue regrowth, and its degradation rate should be synchronized to tissue regrowth. Therefore, at time 0 (as soon as implanted) the graft should mimic the geometry of radial artery. Table 3 shows the features that an electrospun synthetic graft should have to mimic extracellular matrix and radial artery features.

Both the PL- and TB0.2-grafts had and internal diameter of 2.6 mm, in accordance with the diameter of mandrel collector (2.5 mm) covered with aluminium foil, and therefore were within the targeted value range. No shrinkage of the grafts occurred after removal from the mandrel, suggesting that the fibres had deposited properly and dried on the collector. Both PL- and TB0.2-grafts had fibres in the nanometre range on the inner and outer surface. The higher percentage of nanofibers on the outer surface could be explained by a loss of conductivity of the mandrel; the fibres that initially deposited on the mandrel were highly attracted from the collector, but the gradual deposition of more fibres reduced the conductivity of the mandrel and allowed an increase of the flying time and hence of the whipping and stretching of the spinning fibres. Moreover, the comparison of the fibre’s diameter on the outer surface showed a higher amount of nanofibers (98%) in PL-grafts than in TB0.2-grafts (73%). TB0.2-grafts also showed a loss of orientation of the drug loaded fibres. This could be attributed to a presumably higher conductivity of the drug loaded polymer solution, due to the presence of tobramycin’s amino groups, which can carry charge. This higher conductivity can shorten the flying time of the spinning fibres, reducing stretching, and causing electrostatic repulsion, which leads to loss of fibre alignment and orientation [20,23,29]. Concerning porosity, a lower number of pores greater than 5μm on the graft inner surface is desirable to prevent blood leakage, whereas the higher porosity in the outer layer, with greater number of pores having a diameter greater than 5, is needed to allow smooth muscle cells to infiltrate, diffuse into the vascular graft and grow. The higher number of pores greater than 5 μm in the outer layer could allow better smooth muscle cell invasion, and this can be achieved by further decreasing the outer layer fibre diameter and density [27].

The antibacterial assays were performed by immersing the graft samples directly into bacterial suspensions in peptone water, of *S. aureus* and *E. coli*. The peptone water provided a small amount of nourishment to the microorganisms, avoiding excessive stress to the bacterial cells caused by the long test duration, but without boosting their growth, as when complete growth media are used. Moreover, the soy peptides could attach on the surface of the grafts, as autologous proteins can on implanted grafts. The time-kill test was performed over 5 days, because our objective was to prevent early post-operative infections occurring within one week after surgery [8,30,31]. At the same time, an extension of up to 7 days of the antimicrobial assays was not possible due to loss of bacterial viability in the used experimental conditions. The results reported in Table 2 show that, after 24 h, the antibacterial effect was significant for both microorganisms and in the order of magnitude of a 1,000,000-fold reduction for *S. aureus* and a 100,000-fold reduction for *E. coli*. As a comparison, the CEN EN 13697 standard reports that the microbicidal effect of disinfectants should be higher or equal to 4, after 15 min of contact [32]. In this case, because an antibiotic embedded in a polymeric matrix was tested, different time parameters were set and investigated. After 48 h for *E. coli* and 72 h for *S. aureus*, the starting bacterial load (10^7^ CFU/mL) was completely broken down, demonstrating that the TB0.2-grafts were not only able to release the tobramycin in an efficient manner, providing a strong antibacterial effect, but also that the loaded amount of drug was sufficient to allow an extended and effective drug release, within 5 days. Local drug delivery of antibacterial molecules is one of the possible strategies to reduce antibiotic resistance, compared to systemic antibiotic therapy, by simply reducing the amount of administered drug [33]. 

In the assessment of biocompatibility of medical devices, in vitro cytotoxicity tests are the first assay to be performed in order to minimize the use of laboratory animals, as recommended by ISO 10993-2 [34]. These assays can be performed by placing the cells directly in contact with the medical device, or by exposure to the device’s extracts. Several cell types can be used for preliminary cytotoxicity screenings, including murine derived fibroblasts [34,35]. In this work, a preliminary cytotoxicity evaluation on PL and TB0.2 extracts in the same time span used for antimicrobial activity assays was performed in order to acquire comparable data. The rationale was to evaluate possible cytotoxicity which could compromise drug delivery system safety, hence the proposed use of the graft as a drug delivery system. Because a medical device for human use was being investigated, normal human fibroblasts (NHDF) were selected. The PL- and TB0.2-scaffolds showed no cytotoxicity towards NHDF cells up to 5 days. Negative control performed with phenol gave 0% cell viability, whereas the cell viability of all the graft samples was greater than 95%, and it is therefore possible to conclude that both the placebo and the tobramycin loaded graft extracts do not impair cell growth [36]. 

## 4. Materials and Methods

### 4.1. Materials

Poly-L-lactide-co-glycolide (PLGA) Resomer LG 824 S (Lactide:Glycolide 82:18, ester terminated) was purchased from Evonik Röhm GmbH (Evonik Röhm GmbH, Essen, Germany). Dichloromethane (DCM), N, N-dimethylformamide (DMF), Phenol and sodium metabisulphite were supplied from Carlo Erba Reagents (Cornaredo (MI), Italy). Tobramycin, ammonium hydrogen phosphate and tetramethylammonium hydroxide solution (1.0 M ± 0.02 M in H_2_O), PBS tablets (pH 7.4, 0.01 M), thiazolyl blue tetrazolium bromide (MTT, approx.98% TLC), dimethyl sulfoxide (DMSO), trypan blue solution and Dulbecco Modified Eagle’s Medium-high glucose powder (DMEM) were purchased from Merck-Sigma (Merck Life Science S.r.l., Milan, Italy). The 25GA engineered fluid dispensing precision tips for electrospinning were supplied by Nordson EFD (Nordson EFD, Westlake, OH, USA) and 5 mL electrospinning syringes by B. Braun Injekt (B. Braun Injekt, Milan, Italy).

Gamma sterile 0.22 μm 47 mm MCE membranes were purchased from Biosigma (Biosigma, Cona (VE), Italy). Fetal Bovine Serum (FBS, Mycoplasma and Virus secerned), Penicillin-Streptomycin (10× solution) and Trypsin-EDTA (1× in PBS, calcium, magnesium and phenol red free) were purchased from Immunological Science (Immunological Science, Rome, Italy). Tryptone soy broth (TSB) and tryptone soy agar (TSA) were supplied by Oxoid (Oxoid, Basingstoke, UK). Buffered Peptone water and Eugonic broth (with Lecithin, TX-100, polysorbate 80) were purchased from Thermo Fisher (Thermo Fisher, Waltham, MA, USA).

Normal human dermal fibroblast (NHDF) cell line was purchased from Euroclone (Euroclone, Pero (MI), Italy).

MilliQ water (0.067 μS/cm) was obtained from Q-POD equipped with a 0.22 μm Millipak Express 40 filter (Millipore, Milano, Italy).

### 4.2. Methods

#### 4.2.1. Polymer Solution Preparation and Characterisation

A 12.5% *w*/*v* PLGA solution was prepared by dissolving the polymer in 75% *v*/*v* DCM overnight; subsequently, 23.8% *v*/*v* DMF and 1.2% *v*/*v* ultrapure water, containing 0.01% *w*/*v* sodium metabisulphite (Na_2_S_2_O_5_), were added, and the solution was homogenized by magnetic stirring for 90 min. To obtain drug loaded grafts, 0.2% *w*/*w*_PLGA_ tobramycin was dissolved in the water fraction and added directly to the polymer solution. The surface tension of polymer solutions was measured using a Du Noüy ring tensiometer (Krüss GmbH, Hamburg, Germany) at both 25 °C and 35 °C.

The rheological properties of the polymer solutions were assessed with a Malvern Kinexus Pro+ rotational rheometer (NETZSCH-Gerätebau GmbH, Verona, Italy) equipped with a CP4/40 cone geometry; amplitude sweep tests were performed at both 25 °C and 35 °C, at a frequency of 1 Hz and a shear stress rate interval of 0.01 s^−1^–1000 s^−1^, to identify the linear viscoelastic region (LVER) of the polymer solutions. Shear rate ramps were performed within the LVER (at 25 °C and 35 °C) to study the viscosity in dynamic conditions. Viscosity at a share rate of 100 s^−1^ was extrapolated from viscosity versus share rate curves because it represents the viscosity of the solution when subjected to the shear stress exerted by the syringe plunge during electrospinning [19].

#### 4.2.2. Small-Diameter Tubular Grafts Fabrication and Morphological Characterisation

The grafts were produced using the electrospinning technique; a NANON 01A vertical electrospinning set-up (Mecc Co. Instruments, Fukuoka, Japan), equipped with a dehumidifier and a DC power supply was used. The PLGA solution was extruded at a flow rate of 0.3 mL/h, from the spinneret composed of a syringe fitted with a 25-gauge blunt needle. An electric potential of 20 kV was applied. The electrospun fibres were directly collected on a 2.5 mm diameter metal rod, rotating at 3000 rpm and located at 15 cm distance from the spinneret.

The diameter and thickness of the grafts were measured with digital vernier callipers. Samples from the inner and the outer surface of the tubular grafts were fixed on carbon tape and made conductive by vapor phase gold sputtering (11 mA, 120 s). The sputtered samples were observed and imaged using a Zeiss EVO MA10 scanning electron microscope (Carl Zeiss, Oberkochen, Germany). Images were analysed using ImageJ software (version 1.52t, NIH, Maryland, USA) [37].

#### 4.2.3. Drug Content and Encapsulation Efficiency Evaluation of the Electrospun Grafts

Tobramycin loaded electrospun grafts (TB0.2) were completely dissolved in 2 mL of DCM; subsequently, the drug was extracted by adding 3 mL of PBS supplemented with 0.01% *w*/*v* sodium metabisulphite. The organic phase and the aqueous phase were stirred at 1000 rpm for 60 min, and after centrifugation at 6000× *g* for 20 min, the aqueous phase was collected and freeze dried to further proceed with tobramycin quantification. A 96% extraction efficiency was achieved thanks to the use of partial miscible aqueous and organic phases and to the respectively high solubility and insolubility of the drug and PLGA in the aqueous phase.

An HPLC method taken from the literature [38] was adapted for tobramycin quantification. Briefly, chromatography was carried out using a 1260 Infinity HPLC instrument (Agilent Technologies, Santa Clara, CA, USA) equipped with a Column Guard (C8, 4 × 30 mm) and a reverse phase LUNA column (C8(2), 5 μm, 100 Å, 250 × 4.6 mm) from Phenomenex Ltd. (Phenomenex Ltd., Aschaffenburg, Germany). An isocratic mobile phase consisting of a 0.05 M diammonium hydrogen phosphate buffer with pH adjusted to 10.0 using tetramethyl ammonium hydroxide was used at a flow rate of 1 mL/min. The detection was carried out using a UV-Visible detector set at 205 nm. A linear standard curve of tobramycin standards in PBS with 0.01% *w*/*v* sodium metabisulphite, ranging in concentration from 50 μg/mL to 900 μg/mL, was calculated (R^2^ = 0.995).

The total drug content (DC) of tobramycin loaded grafts expressed in μg/mg and the encapsulation efficiency (EE%) of the drug into the electrospun fibres were calculated according to Equations (1) and (2), respectively:DC = M_Tex_/M_graft_(1)
where M_Tex_ is the mass of tobramycin (μg) actually extracted from the graft and M_graft_ is the mass of the graft (mg) from which the drug was recovered.
EE% = (DC/M_Ttheor_) × 100(2)
where DC is the amount of tobramycin (μg) in the graft and M_Ttheor_ is the amount of tobramycin (μg) added to the polymer solution (theoretically loaded) used to produce the graft.

#### 4.2.4. Antimicrobic Activity Evaluation

Time-kill assays [39] were performed on two representative bacterial strains for bacterial infections, *Staphylococcus aureus* ATCC 6538 and *Escherichia coli* ATCC 10536, to evaluate the microbicidal effect (ME) of the tobramycin loaded grafts over a time frame of 5 days. 

The bacteria were cultured overnight at 37 °C in Tryptone Soy Broth (TSB). The obtained cultures were centrifuged (2000 rpm for 25 min), the supernatant was removed and the microorganism pellet was resuspended in sterile water. The optical density of the microbial suspensions was adjusted to A = 0.3 (wavelength 650 nm), which corresponds to a bacterial titre of 1 × 10^7^–10^8^ CFU/mL (Colony Forming Units per millilitre). The final testing bacterial suspensions were prepared by diluting 1 mL of each A = 0.3 microbial suspension in 4 mL of buffered peptone water. 

The tested tobramycin grafts (TB0.2) were cut to obtain a 3 cm length piece and sanitized under UV light overnight; the same was done with the placebo vascular grafts (PL), which were used as a positive microbial growth control. Each sample (TB0.2 and PL) was immersed in a testing bacterial suspension and incubated at 37 °C in aerobic conditions. At fixed times (24, 48, 72, 96 and 120 h) a 50 μL aliquot of the bacterial suspension was collected and serially diluted in an antibiotic neutralizing agent (Eugonic broth). An amount of 1 mL of the 1:10^2^, 1:10^4^, 1:10^6^ and 1:10^7^ dilutions were inoculated into Tryptone Soy Agar (TSA), and the plates were incubated for 24 h at 37 °C. After incubation, the bacterial colonies were enumerated, and viable bacterial titres were calculated after contact with TB0.2 and PL and expressed as CFU/mL. The ME at each time of the tobramycin loaded grafts was calculated using Equation (3) [40]:
ME = log_10_ NPL − log_10_ NTB0.2(3)
where NPL is the viable bacterial titre in CFU/mL after contact with the placebo graft and NTB0.2 is the viable bacterial titre in CFU/mL after contact with the tobramycin loaded graft.

#### 4.2.5. Cytotoxicity Evaluation

An indirect cytotoxicity test (ISO 10993-5-2009, [36]) was performed using thiazolyl blue tetrazolium bromide (MTT) assay [34]; both PL and TB0.2 samples with equal weight were incubated in DMEM supplemented with 10% FBS and 1% Penicillin-Streptomycin. The PH of the incubation medium was measured along incubation (827 pH lab pH-meter, Methron ion analysis, Switzerland), in order to evaluate possible pH shifts. The extracts obtained at 24, 48, 72, 96 and 120 h were incubated with previously seeded NHDF (P-9; 10,000 cells/well) for 24 h at 37 °C, 5% CO_2_ and 90% relative humidity; a sample of cells kept in culture without any addiction and undergoing the same protocol, was the test control (Ctr). After the set incubation time, the extracts were removed and the cells were gently washed with sterile PBS; the MTT solution in DMEM (0.55 mg/mL) was added and incubated for 150 min at 37 °C. Phenol was used as the negative cell viability control. The MTT solution was removed, and 200 μL DMSO was added to dissolve the formazan crystals before absorbance readings were taken at 570 nm using a MPP-96 HiPo Microplate photometer (Biosan, Riga, Latvia). Cell viability (CV%) was calculated according to the following Equation (4):CV% = (ABS_T_ − ABS_Bl_) × 100/(ABS_Co_ − ABS_Bl_)(4)
where ABS_T_ in the absorbance of the extract threated group, ABS_Co_ is the absorbance of the control group and ABS_Bl_ is the absorbance of the blank (DMSO).

#### 4.2.6. Statistical Analyses

Statistical analyses were performed using GraphPad Prism 9.0 software (San Diego, CA, USA). A 2-sample *t*-test was performed to determine significance between two experimental groups, and single-factor analysis of variance (ANOVA) with Tukey’s multiple comparison test was performed to find statistical differences between three or more experimental groups. Simple correlation analysis was performed to determine significance in correlation between two factors. Statistical significance is represented as * for *p* < 0.05, ** for *p* < 0.01, *** for *p* < 0.005 and ns for non-significant.

## 5. Conclusions

In this preliminary work, a small-diameter biodegradable vascular grafts prototype loaded with tobramycin was designed and manufactured. The Electrospinning technique, together with the apparatus outfit, were suitably studied in order to obtain fibres with morphological features close to those of the extracellular matrix, as far as fibre diameter and pore size is concerned, making the prototype suitable for tissue engineering applications. Moreover, the inner diameter and thickness of the tubular scaffold were designed, and they were achieved, conforming to radial artery geometry. The tubular graft released an effective tobramycin concentration and ensured a strong antibacterial activity over 5 days, making it a promising local drug delivery device. The paper provides a detailed discussion of electrospinning process parameter selection, and their rationale, which could be helpful for the researchers in the field. The tubular graft characterization needs to be implemented by evaluation of their mechanical properties and their behaviour under flow and pressure in vitro tests, and by biological tests on graft thrombogenicity. In these terms, the positive results obtained in this preliminary research will be the starting point for further in vitro and in vivo experiments aimed to obtain an implantable synthetic graft for small vessel regeneration.

## Figures and Tables

**Figure 1 ijms-22-13557-f001:**
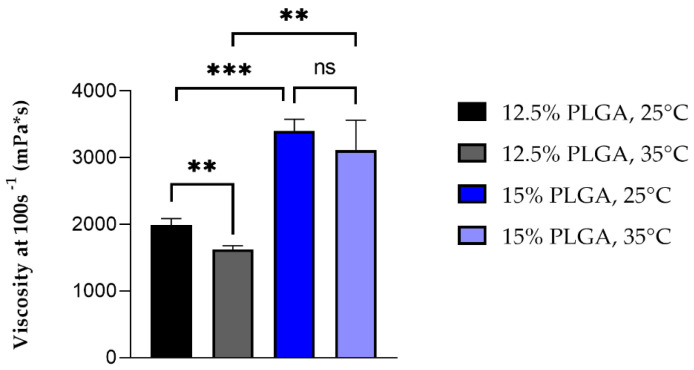
Average viscosity of 12.5% and 15% *w*/*v* PLGA solutions at 25 °C and 35 °C, (*n* = 3, ** stands for *p* < 0.01, *** for *p* < 0.005 and ns for non-significant).

**Figure 2 ijms-22-13557-f002:**
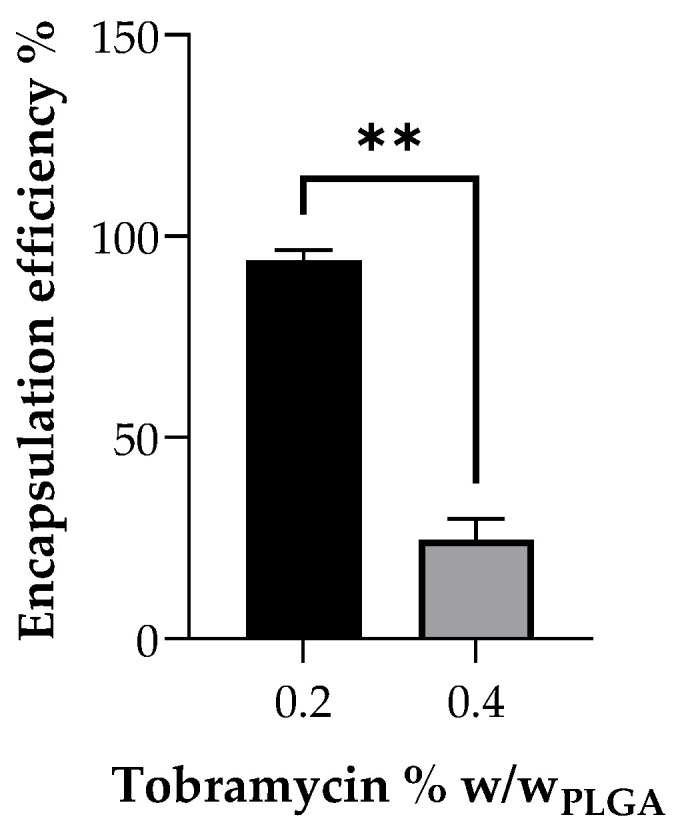
Encapsulation efficiency of tobramycin into electrospun fibres, depending on the loaded amount of the drug into the polymer solution before spinning (*n* = 3, ** stands for *p* < 0.01).

**Figure 3 ijms-22-13557-f003:**
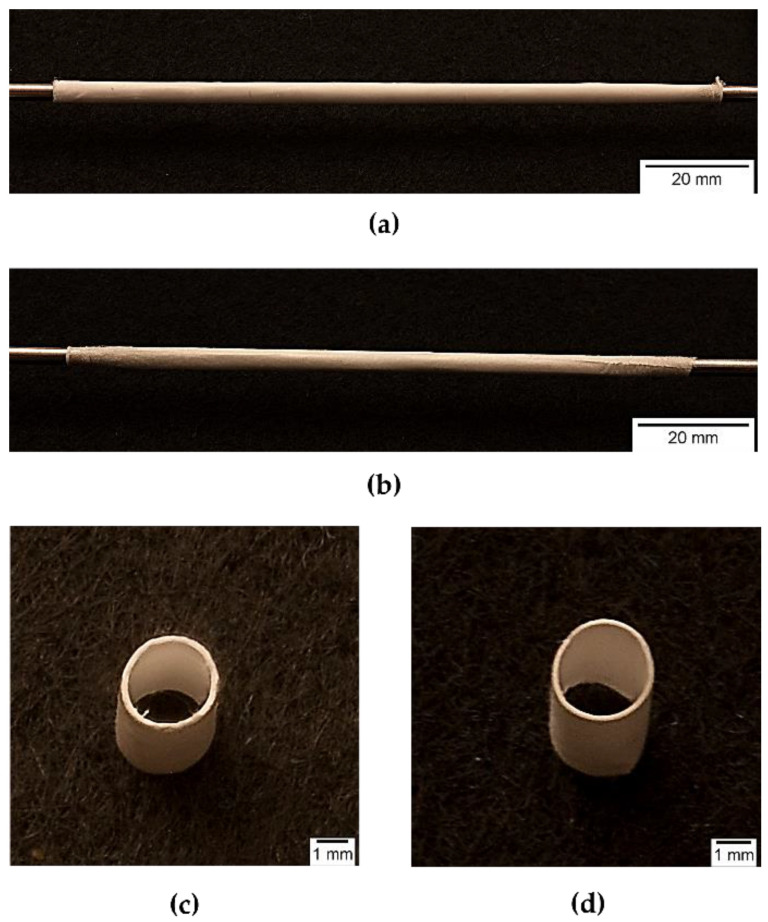
Photographs of electrospun vascular grafts: whole (**a**) PL-graft and (**b**) TB0.2-graft on collector; (**c**) PL-graft and (**d**) TB0.2 graft horizontal sections.

**Figure 4 ijms-22-13557-f004:**
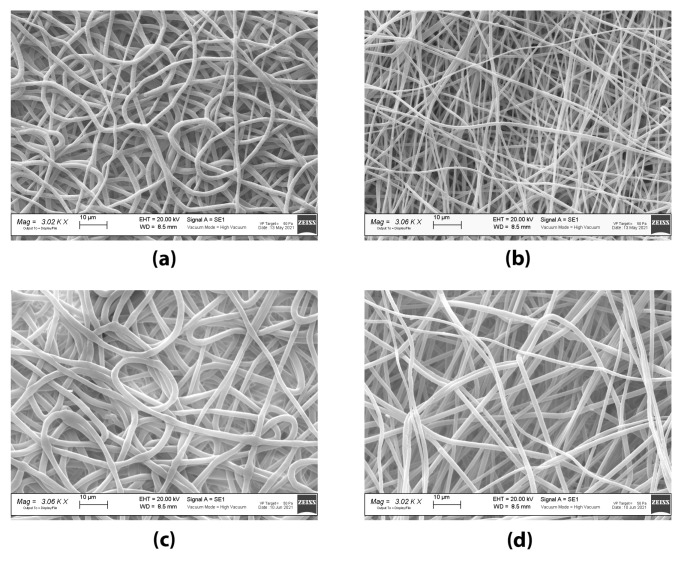
SEM images of electrospun small-diameter vascular grafts: (**a**) inner lumen surface of PL-graft; (**b**) outer surface of PL-graft; (**c**) inner lumen surface of TB0.2-graft; (**d**) outer surface of TB0.2-graft.

**Figure 5 ijms-22-13557-f005:**
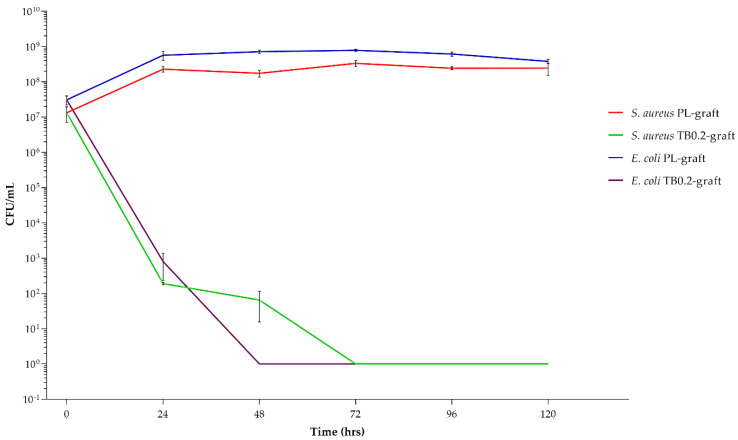
Bacterial population expressed as CFU/mL of *S. aureus* and *E. coli*, when placed in contact with placebo (PL-grafts) and tobramycin loaded grafts (TB0.2-grafts) for 5 days.

**Figure 6 ijms-22-13557-f006:**
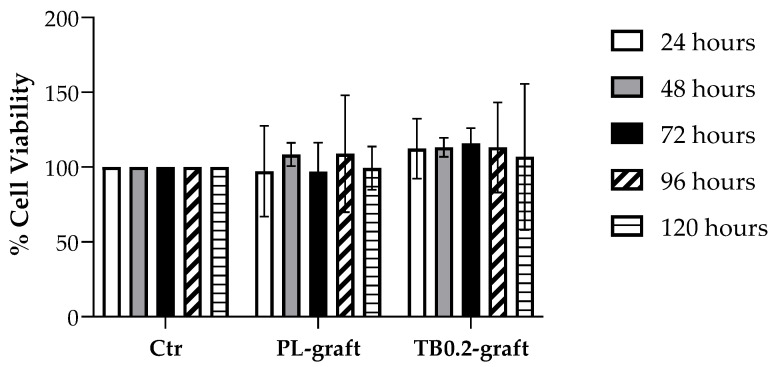
Cell viability expressed as % of viable NHDF after contact with extracts of placebo (PL-graft) and tobramycin loaded grafts (TB0.2-graft), in comparison with the 100% cell viability of the control (Ctr).

**Table 1 ijms-22-13557-t001:** Viscosity and surface tension values of PLGA solutions (75% *w*/*v* DCM: 23.8% *w*/*v* DMF: 1.2% *w*/*v* Na_2_S_2_O_5_ water solution) at 25 °C and 35 °C (Mean ± standard deviation, *n* = 3).

PLGA Concentration (% *w*/*v*)	Temperature (°C)	Viscosity at 100 s^−1^ (mPa*s)	Surface Tension (dyne/cm)
12	25	1990 ± 93.5	33.86 ± 0.26
	35	1620 ± 59.9	33.62 ± 0.61
15	25	3402 ± 173.6	35.09 ± 0.61
	35	3116 ± 445.7	36.23 ± 1.80

**Table 2 ijms-22-13557-t002:** Microbicidal effect (ME) of TB0.2-grafts on *S. aureus* and *E. coli*, at a time of 5 days. (ME ± standard deviation, *n* = 3).

Time	ME	ME
(h)	*S. aureus*	*E. coli*
24	6.08 ± 0.11	5.90 ± 0.46
48	6.50 ± 0.28	8.85 ± 0.05
72	8.52 ± 0.09	8.89 ± 0.03
96	8.38 ± 0.04	8.79 ± 0.06
120	8.37 ± 0.17	8.58 ± 0.06

**Table 3 ijms-22-13557-t003:** Extracellular matrix and radial artery morphologic properties and related target value, for the design of the electrospun vascular graft.

Extracellular Matrix	Target Values	Ref.
Fibre diameter	<1 μm	[27]
Pore size inner layer	<5 μm	
Pore size outer layer	>5 μm, interconnecting pores	
**Radial artery**		
Internal diameter	2.56–2.77 mm	[28]
Wall thickness	220–350 μm	

## Data Availability

The data presented in this study are available in the article.

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
