# Peer review of "Tobramycin Supplemented Small-Diameter Vascular Grafts for Local Antibiotic Delivery: A Preliminary Formulation Study"

_ijms, 2021, doi:10.3390/ijms222413557_

Round 1

Reviewer 1 Report

Comments:

  1. It is confusing of why the aim of this work is to design a 2.5 mm graft? Is it the standard size of current off-the-shelf grafts for peripheral bypass procedures? Typically, synthetic grafts smaller than 5 mm are more prone to occlude due to thrombosis and a lack of functional endothelium rather than infection.
  2. Scale bars should be added for Figure 3.
  3.  Add more references to sentence 305: "The time-kill test was performed....one week after surgery"
  4. Can you comment on the thrombogenicity of your graft?  
  5. Please include and state the limitations of your study.
  6.  Please include justification for the use of human dermal fibroblasts in your cell viability study. Why was no negative control utilized in this portion of the study?
  7. The authors state that their grafts were designed to mimic properties of a radial artery, however no mechanical properties were presented?  The authors need to clarify their statement. 
  8. Please add standard deviation bars to figure 5.
  9. The overall tone of the manuscript should be lowered. In the conclusion statement, the word 'stable' is mentioned, however, this graft was only tested in vitro and without flow or pressure. Also, the sentence 'it is therefore suitable as a local drug delivery device' is an overreach. Again, this was very simplistic testing an in vitro model, with no flow or pressure.
  10. If this graft is intended to be biodegradable, why are the authors choosing to mimic the geometry of the non-degradable radial artery? Shouldn't the thickness/geometry of your artery be decided by degradation rate?

Author Response

Reviewer 1

Open Review

Comments:

  1. It is confusing of why the aim of this work is to design a 2.5 mm graft? Is it the standard size of current off-the-shelf grafts for peripheral bypass procedures? Typically, synthetic grafts smaller than 5 mm are more prone to occlude due to thrombosis and a lack of functional endothelium rather than infection.

AUTHORS ANSWER: the authors thank reviewer comment, which helps in clarifying the goal of the work. The experimental work carried out is focused on designing 2.5 mm grafts for  surgical treatment of peripheral artery occlusive disease (PAOD). In the manuscript introduction the authors explained what is PAOD, which are the patients most affected by this pathology and which is the therapy for this pathology. The authors are aware that 2.5 mm graft is not the standard size of current off-the-shelf grafts, and that graft with diameter lower than 5 mm are more prone to occlude due to thrombosis and a lack of functional endothelium. Moreover, as the authors report in the paper introduction, about 1% -5% of the patients develop infections after implantation of vascular prosthetics. For this reason, in this preliminary work, the authors focus is on designing a scaffold supplemented with antibiotic that could act as drug delivery system. The authors have better detailed in the introduction of the revised version of the paper.

  1. Scale bars should be added for Figure 3.

AUTHOR ANSWER: The authors thanks for the reviewer’s observation and apologize for uploading images with missing scale bar. New images will be provided, including appropriate scale bar.

  1. Add more references to sentence 305: "The time-kill test was performed....one week after surgery"

 AUTHORS ANSWER: the authors agree with reviewer comment and added more references in the revised version of the paper in order corroborate sentence of line 305, namely:

Kilic A., Arnaoutakis D. J., Reifsnyder T., Black III J.H, Abularrage C. J., Perler B.A. and Lum Y. W., Management of infected vascular grafts. Vascular Medicine 2016, Vol. 21(1) 53–60.

Anagnostopoulos A., Ledergerber B., Kuster S.P., Scherrer A. U., Näf B., Greiner M. A., Rancic Z., Kobe A., Bettex D., Hasse B., Inadequate Perioperative Prophylaxis and Postsurgical Complications After Graft Implantation Are Important Risk Factors for Subsequent Vascular Graft Infections: Prospective Results From the Vascular Graft Infection Cohort Study. Clinical Infectious Diseases, 2018, 621-630.

  1. Can you comment on the thrombogenicity of your graft?

AUTHOR ANSWER: the authors thank reviewer for the comment that is relevant to the topic. In this preliminary work the authors focused on formulation study and on evaluating tobramycin antimicrobial activity; graft thrombogenicity was not evaluated. The authors modify the manuscript test introduction according to reviewer comment, also considering the comment at following point 9 and in order to clarify the work is a preliminary one, its main focus and limitations.

  1. Please include and state the limitations of your study.

AUTHOR ANSWER: according to reviewer comment the authors included in Conclusion section of the revised version of the manuscript, a statement about the limitations of their study. They highlighted that the tubular grafts characterization needs to be implemented by evaluation of their mechanical properties and their behavior under flow and pressure in vitro tests and also by biological tests on graft thrombogenicity. Indeed, the authors underlined that the paper provides a detailed discussion of electrospinning process parameters selection, and their rationale, that could be helpful for the researchers in the field.

  1. Please include justification for the use of human dermal fibroblasts in your cell viability study. Why was no negative control utilized in this portion of the study?

AUTHOR ANSWER: Following the ISO 10993 (Ref.36), Ref.34 (Goud, N.S. Biocompatibility Evaluation of Medical Devices. In A Comprehensive Guide to Toxicology in Nonclinical Drug Development; 2017; pp. 825–840 ISBN 9780128036204) and Ref.35 (Melisande, B.; Emile, J.; D., P. michael; Najet, Y. Biocompatibility of polymer-based biomaterials and medical devices – regulations, in vitro screening and risk-management, Biomater. Sci. 2018, 6, doi:10.1039/C8BM00518D), that were added in the revised manuscript by the authors, murine fibroblasts are used model cells to preliminary test medical device cytotoxicity. Since the authors are investigating a medical device for human use, they selected human fibroblasts.

Phenol was used as negative control, giving a cell viability equal to 0%. The authors did not mention the use of a negative control, because they focus on the outcomes of the indirect cell toxicity test on NHDF. The authors will modify the Methods and Material section of the article, including the missing information about the performance of a negative control.  

  1. The authors state that their grafts were designed to mimic properties of a radial artery, however no mechanical properties were presented? The authors need to clarify their statement. 

AUTHOR ANSWER: the authors are aware that mechanical properties are an important point in the characterization of a vascular graft. They took in consideration the relevant comment of reviewer reporting their explanation here below.

In this preliminary work the authors focused on the features reported in Table 3 of the manuscript as those required for an electrospun synthetic graft to mimic extracellular matrix and radial artery.

Extracellular matrix

Target values

Ref.

 Fiber diameter

< 1 μm

[27]

 Pore size inner layer

< 5μm

 Pore size outer layer

> 5μm, interconnecting pores

 Radial artery

 Internal diameter

2.56 – 2.77 mm

[28]

 Wall thickness

220 – 350 μm

In the Discussion section of the manuscript the authors extensively discussed these properties together with polymer and electrospinning process parameters rationale of selection in order to achieve them. The authors hope they sufficiently clarified the meaning of the statement “designed to mimic properties of a radial artery”.

Moreover, the authors are going to study the vascular grafts mechanical properties in their next work, relating them to the vascular graft degradation behavior.

  1. Please add standard deviation bars to figure 5.

AUTHOR ANSWER: The authors thank the reviewer for his/her observation and precise that standard deviation bars were already included in figure number 5, but standard deviations are verry small and barely visible because of the logarithmic scale. We will include a modified figure with more visible standard deviation bars, but we would like to point out that, in our opinion, the new figure is less intelligible. 

  1. The overall tone of the manuscript should be lowered. In the conclusion statement, the word 'stable' is mentioned, however, this graft was only tested in vitro and without flow or pressure. Also, the sentence 'it is therefore suitable as a local drug delivery device' is an overreach. Again, this was very simplistic testing an in vitro model, with no flow or pressure.

 AUTHOR ANSWER: the authors agree with reviewer comment and thoroughly revised the whole manuscript accordingly. In particular, conclusions have been completely rewritten in the revised version of the manuscript, in order to comply with the reviewer comment.

  1. If this graft is intended to be biodegradable, why are the authors choosing to mimic the geometry of the non-degradable radial artery? Shouldn't the thickness/geometry of your artery be decided by degradation rate?

AUTHOR ANSWER: the authors thank for this relevant comment and give the following explanation for choosing to mimic the geometry of the non-degradable radial artery. The graft should mimic the geometry of the non-degradable radial artery because it is intended to be implanted to replace it. The rationale is that the graft should support tissue regrowth and its degradation rate should be synchronized to tissue regrowth. Therefore, at time 0 (as soon as implanted) the graft should mimic the geometry of radial artery. The explanation has been added into the revised version of the manuscript with the aim to contribute to clarify the rationale of their work.

Reviewer 2 Report

The creation of antimicrobial protection for vascular prostheses is an important task. The authors propose adding tobramycin directly into Poly-L-lactic-co-glycolic acid (PLGA) solution in order to encapsulate the drug into the electrospun fibres.

Noted, incapsulating tobramycin into PLGA polymer induced change of polymer properties (viscosity and surface tension) and scaffold microstructure.  Mechanical strength of the prosthesis is a significant characteristic. Weak tensile strength and stiffness of vascular prosthesis make the existing good antimicrobial properties useless.

The indirect MTT-test include the influent of the cultural medium after prostheses cultivation on the fibroblast viability. This approach is not enough sensitive to assess the cytotoxicity of the material. It is desirable direct cell seeding (form of polymer films, fix them to the plate bottom and made cell seeds). In addition, the choice of short time culturing (24 hours) for stress-resistant fibroblasts do not allow to get objective assessment cytotoxicity of the material.

Figure 6. The results must be presented within 100% of the viability scale. There is no information about control. The PLGA degradation lead to increase of acidity into environment and tobramycin possibility toxic to cells. Why did the samples (Pl-graft and Tb0.2-graft) have 150-200% viability compared to control? Does Pl-graft and Tb0.2-graft media really stimulate cell viability?

Author Response

Point by point answer to reviewer 2, are atacched as pdf file ANSWER 3

Round 2

Reviewer 2 Report

Dear authors, Thank you for the work that you have done.

You corrections and clarifications have provided a better understanding of the purpose and results of investigation.